# The Feasibility of Remotely Piloted Aircrafts for VOR Flight Inspection

**DOI:** 10.3390/s20071947

**Published:** 2020-03-31

**Authors:** Diogo de Oliveira Costa, Neusa Maria Franco Oliveira, Roberto d’Amore

**Affiliations:** 1Mechanical Engineering Department, Universidade de Brasília, UnB-Brasília, Brasília DF 70910-900, Brazil; 2Instituto Tecnológico de Aeronáutica, IEE, Pça. Mal. Eduardo Gomes 50, São José dos Campos 12228-900, Brazil; neusa@ita.br (N.M.F.O.); damore@ita.br (R.d.A.)

**Keywords:** flight inspection system, NavAid, VOR, ICAO, unmanned aerial systems, RPA, UAV, drones

## Abstract

This article analyzes the use of Remotely Piloted Aircrafts (RPA) in VOR (Very High Frequency Omnidirectional Range) flight inspection. Initially, tests were performed to check whether the Autopilot Positioning System (APS) met the regulatory requirements. The results of these tests indicated that the APS provided information within the standard regulations. A Hardware in the Loop (HIL) platform was implemented to perform flight tests following the waypoints generated by a mission automation routine. One test was performed without introducing disturbance into the proposed test platform. The other four tests were performed introducing errors in latitude and longitude in the APS into the platform. The errors introduced had the same characteristics as those measured in the initial tests, in order for the simulation tests to be as similar as possible to the real situation. The tests performed with positioning errors only did not lead to false misalignment detection. However, introducing positioning errors and a 4° VOR misalignment error, a misalignment of 3.99° was observed during the flight test. This is a value greater than the maximum one allowed by the regulations, and the system indicates the VOR misalignment. Five flight inspection tests were performed. In addition to the APS errors, tests with a modulation error were also conducted. Introducing a 4° VOR misalignment in conjunction with modulation error, a misalignment of 4.02° was observed, resulting in successful misalignment detection.

## 1. Introduction

Radio Navigation Aids (NavAids) provide electronic guidance for aircraft, helping their navigation in poor visual conditions, which improves aviation safety in the context of the ever-increasing demand for air traffic [1,2]. NavAids must pass periodical inspections known as flight inspections to ensure their calibration and air system safety. The International Civil Organization (ICAO) specifies the inspection periodicity and how these inspections must be made [3]. Due to the periodic repetition and the high cost of the flight mission inspection [4], the interest in alternatives for performing these inspections has arisen. VOR (Very High Frequency Omnidirectional Range) is one of the most common types of NavAid used in commercial aviation, and its inspection requires the displacement of a well-equipped aircraft to the corresponding airport. 

The improvement of the technological resources and the popularization of Remotely Piloted Aircrafts (RPA) make it possible to increase the complexly of the missions. Readily available Auto-Pilot systems (AP) allow complex trajectories with Way-Points (WP), which have not been possible before [5].

Unmanned Aircraft (UA) is a term that encompasses different kinds of aircraft, like Unmanned Aerial Vehicles (UAVs), Drones, and RPAs. We consider an RPA to be an UA that can be integrated into the same airspace as manned aircraft, and which is subject to the same certification requirements as a manned aircraft [6]. When referring to others works, we use the same denominations adopted by the authors. 

In accordance with regulations, the VOR system must be checked periodically by means of inspection flights. At small aerodromes, the air traffic is restricted, except for flight inspection aircraft. At larger airports, flight inspections are performed by flights inserted in the normal air traffic. In both cases, the Air Traffic Controller (ATCo) of the region is aware of the special operations required for the flight inspections [5]. 

There are many types of airspace restrictions related to the use of RPAs due to the lack of flight safety and reliability in these devices [7,8]. As flight inspections also need special air traffic operations, the use of RPAs in these flight missions is an option that should be considered. The potential of UA was foreseen in a work discussing the future of flight inspections [9]. Due to the level of technological maturity in 2006, the author pointed out some issues that must be considered before the proposal becomes viable. The main concerns raised were: certification requirements regarding the maximum error; the cost of a ground pilot to supervise; and the cost of an Unmanned Aerial Vehicle (UAV) with certification for civil airspace. A first attempt at using UAs in flight inspection was proposed in [10], in which the architecture of the avionics to be embedded in an UAV were described, aiming towards the optimization of flight inspection with respect to the platform and operating costs, increasing the availability of the system for flight inspection. In a further development presented in [11], the proposed architecture was refined and detailed, presenting the equipment to be used to implement the architecture proposed. The system requirements, cost estimation, and equipment weight were also discussed. Simulations using a commercial autopilot were performed, and the results indicated that flight inspection procedures could be executed by an RPA. Reference [5] deals with the problem of waypoints generation for flight simulations. An XML (eXtensible Markup Language)-based specification language to define complex RPA missions was proposed. The flight path to a specific waypoint was established by “legs”, and several primitives for a “leg” were provided. Reference [12] described the use of an UA for corrective and preventive maintenance of an Instrument Landing System (ILS). A 3.7 kg commercial flight analyzer, capable of performing measurements on ILS and VOR, was installed in an UAV. Good results were reported regarding the repeatability of measurements due to the accuracy of the UAV in following a defined trajectory. However, the use of the system for VOR inspections will be explored in a further development. In Ref. [13], a flight inspection system using an UA with a 50 kg payload capacity was presented. A ground station was responsible for controlling the UAV’s maneuvers with a 100 km data link coverage. Preliminary tests were performed, but no inspection flight tests were reported. This paper presented the infrastructure available, the aerial platform characteristics, and the embedded avionics. Dedicated sensors for flight inspection VOR/LOC and DME were embedded in the aerial platform, among others related, for example, to ground surveillance. Reference [14] and its previous work, [15], deal with the payload (antennas and receivers) designed to measure Radio Frequency (RF) requirements for ILS. Two octocopters were used to demonstrate the application of the proposed system. These platforms allow quasi-stationary hovering, and this paper was focused on the new measurement capabilities for the signal-in-space of conventional Navaids and some fundamental prerequisites for signal analysis on UA platforms in a near-ground environment. 

The use of UAs, capable of flying with flight test equipment, has to comply with regulatory requirements defined by government agencies. RF standards, quality acceptance tests, safety monitoring, and the integration of UAs in airspace are questions that need to be addressed [8,16,17]. Regarding the integration of UAs in airspace, due to the increase in the maturity of UA systems, the use of UA flights in controlled airspaces has been reported without segregation of the airspace [18]. The UA, connected to a mobile network, transmits its position to a traffic management system that processes the UA position while at the same time showing the UA with the tracking data of manned aircraft. The development of this system was possible thanks to a research project between *Deutsche Flugsicherung* (DFS), the company responsible for controlling the air traffic in Germany, and *Deutsche Telekom*, a mobile communications provider [19]. 

VOR inspection flights should evaluate whether the ground equipment meets the requirements defined by ICAO. However, the current state of research for VOR inspections using RPA has not given proper attention to ICAO regulations. This work addresses the problems of executing VOR inspections using RPA, considering ICAO regulations to be a relevant issue. A HIL (Hardware in the Loop) platform using a commercial open code autopilot to control a simulated RPA was developed to test the feasibility of the approach. The implemented HIL, which integrates the AP hardware and the flight simulator, allows the simulated aircraft to be controlled by a radio controller or by an AP. 

A routine for automatic generation of a standard inspection trajectory was developed. This routine minimizes the human interaction in order to ensure compliance with regulatory rules. The routine generates all the WPs that are to be used throughout the mission, according to input parameters like airport, cruise velocity, and GPS acquisition rate. Therefore, different missions can be planned in a standardized way.

Experiments were performed in order to verify whether the Autopilot Positioning System (APS) has the sample rate, precision and accuracy that are necessary to verify that the VOR system meets the regulatory norms. Experiments were also performed to check whether the APS complies with ICAO regulations. The APS is employed by the AP to determine the geographic position by combining GPS and Inertial Navigation System (INS) data.

The rest of this paper is organized as follows. Section 2 discusses the VOR operation and presents a flowchart showing the approach for assessing the feasibility of the approach. Section 3 evaluates the Autopilot Positioning characteristics by means of a series of experiments. In this section, it is also analyzed whether the positioning system is within the tolerance range established by the ICAO regulations. Section 4 presents the HIL platform and a routine for generating inspection trajectories. Section 5 analyzes five flight inspections performed by HIL simulations, and discusses the results considering the ICAO regulations.

## 2. Materials and Methods

VOR indicates the magnetic bearing from the station to the aircraft enabling an aircraft to stay on course by means of radio signals transmitted by a network of fixed ground radio beacons. The VOR ground station sends out an omnidirectional master signal, and a highly directional second signal is propagated by a phased antenna array rotating clockwise at 30 Hz. 

VOR equipment emits signals which carry the north reference. Depending on the position of a body in the space, receiving the VOR signal, this body receives the information of its position related to the north reference, see Figure 1. The space around the VOR is divided by 360 radials, given a resolution of 1°, with each one carrying information about its angular orientation in relation to the North magnetic [20]. Using the aeronautical nomenclature for Navigation Aids equipment, the radials are numbered by their angles in reference to the North magnetic, in a clockwise direction [21]. In this way, the position of an airplane can be determined, in relation to the North magnetic, through the VOR signal. It is important to note that the information given by the VOR, which is the radial related to the magnetic north that the aircraft is crossing during the flight, is independent of the aircraft heading, i.e., the direction in which the aircraft is flying. For example, Figure 1 shows two aircraft and a VOR station. Aircraft-1 is in radial 315 and Aircraft-2 in radial 90. It also shows that the heading of the aircraft is not always coincident with the radial. The heading of Aircraft-1 is 45°, while Aircraft-2 is aligned with the radial 90°. As depicted in Figure 1, an aircraft crossing the VOR radial 315° may have a flight direction (given by the airplane heading) in other direction, 45°, for example. An aircraft can also have a heading of 0° (a north heading), but is in the radial 45°, for example. Or, with the same heading, it can be in radial 190°, for instance.

A VOR inspection shall consider alignment errors and three modulation errors: bend, roughness and scalloping [3]. The modulation errors, bend, roughness and scalloping, occur due to geographic factors, as region topography, mobile towers or edifications. These errors, also called structural errors, affect the signal quality and, depending on their intensity, they can make some radials non navigable. These errors factors are complex and must be considered [22].

A VOR inspection consists of comparing the VOR radial information in which the aircraft is flying, with the radial position of the aircraft, which will be used as reference. The radial position of the aircraft is computed using its position and the VOR station position. Thus, the radial position of the aircraft must be computed periodically, which means the radial position of the aircraft must be read in an established sample rate. Therefore, this sample rate and the RPA velocity, among other things, are parameters impacting in the viability of the proposed solution.

To use the computed radial position as a reference in the inspection, it is necessary to know the precision and accuracy of the aircraft’s positioning system. In this way, the quality of the positioning information provided by the aircraft positioning system must be checked to verify whether it meets the regulation requirements. 

The guidance on the VOR flight inspection by the ICAO regulation is defined in [3]. It is established that the alignment can be determined by flying an orbit or by flying a series of radials. The orbit should have enough overlaps to ensure that the measurement covers all 360 radials. By this regulation, the alignment of the VOR is determined by averaging the error throughout the orbit, or by flying a series of radial approach. The alignment pattern accuracy deviation has a tolerance of ±2° and an uncertainty of 0.6°. 

The regulation establishes the pattern accuracy due to modulation errors (bend, roughness and scalloping), but these are not the main target of these study. However, some results considering these modulation errors are presented. 

In terms of the height in which the mission should be performed, it is established by regulations [23] a height within the limits, between 4° and 6°. 

As an example, using these data regulations, it is possible to verify some trajectory parameters and their correspondence with the regulation limits. In a circular inspection trajectory of 0.5 NM (Nautical Miles), i.e., 926 m radius, a 0.6° uncertainty corresponds to 9.6 m. Therefore, to meet the ICAO regulations, the positioning system to be used in VOR inspection must provide position data with error lower than 9.6 m. If this flight is conducted at 90 m above ground, this corresponds to an elevation angle of 5.5° to the chosen mission radius, which is within the limits between 4° and 6°.

### Global Approach for Assessing Feasibility

Figure 2 shows the global approach for assessing the feasibility. Section 3 analyzes the APS’s errors in latitude and longitude data to check if the APS meets the ICAO requirements for a VOR inspection. Two experiments were performed, and a correction system was proposed to reduce the static error. In Section 4, the HIL platform is presented, and the parameters for a flight mission are analyzed. In addition, a routine for automatic generation of inspection trajectory is described, and two examples of inspection trajectories in commercial airports are depicted. The radius of the circular trajectory is the main parameter, and its impact on detecting misalignment errors, is analyzed. As the misalignment due to APS’s errors can be reduced by increasing number of flights around the VOR, a trade-off analysis was made through flight tests. In Section 5, the HIL Processing block is explained in detail, and five flight tests are performed using the HIL platform. No errors were added in *Test 1*, and in *Test 2* only the APS’s errors were added. In these tests, the missions should not detect any problem with the VOR, even with the APS error. In *Test 3*, *Test 4* and *Test 5*, VOR alignment and modulation errors were added as depicted in Figure 2. In all of these tests, the misalignment information resulting from the system inspection should be proportional to the errors added and in accordance with the regulations.

## 3. Autopilot Positioning System Characteristics

Experiments were conducted to verify whether the aircraft position in the flight mission, indicated by APS, was within the tolerance range established by regulations. For this, it is necessary to analyze the errors in the APS data. The available APS used in this work integrates GPS and INS. 

### 3.1. APS Experimental Evaluation

Two types of experimental trials on the APS were carried out: a static one and a dynamic one. The static experiment checked the systematic error (the tendency to produce results that differ in a systematic manner from the true values) in the latitude and longitude information provided by the APS. The dynamic experiment checked the distance traveled by the aircraft during the time interval necessary for the APS to acquire two consecutive samples, and how this distance was related to the APS sample rate and the RPA velocity.

#### 3.1.1. Static Experiment (Using Geodetic Mark)

To verify and quantify the systematic error in the information provided by the APS, the system was settled in two geodetic marks available at SJK airport: mark 1 at (23°13′22.616″ S, 45°51′51419″ W), and mark 2 at (23°13′20.607″ S, 45°51′52.815″ W). In Brazil, where the SJK airport is located, the legal geodesic reference systems are SAD69 (South American Datum 1969) and the SIRGAS2000 (Geocentric reference system for Americas). Information regarding the samples’ positions was taken for 40 min for each position, and the median values of the samples were compared with the reference position value. This period of time for data acquisition was arbitrarily chosen, but it was chosen with aim of providing statistically significant information regarding the mean position given by the GPS and its standard deviation. The sample time in the data acquisition was 7 Hz. Figure 3 depicts the results of this experiment, and Table 1 presents the mean error value, *ϵ_mean_*, and standard deviation, *σm*, sampled at each geodetic mark. The mean error value was computed taking the mean position computed from all the positions sampled in each experiment, and then the error was calculated from this mean position and the mark position. It is possible to see the distance between the mean position and the mark position. The standard deviation was computed using the computed mean position and all the samples in each experiment. The mean error and the standard deviation in position throughout this work were computed in this way, unless stated otherwise. In Figure 3, a circle with a radius of 1 m is presented as a reference for the error and dispersion in the samples. The acceptable position uncertainty defined by the regulations, as described previously, depends on the parameters of the inspection mission. Considering the mission example presented in Section 2, for a mission inspection trajectory with a 0.5 NM (926 m) radius, the acceptable 0.6° uncertainty corresponds to 9.6 m.

The higher mean error observed for the data sampled at geodetic mark 1, Figure 3a, may be explained by the interference of a steel hangar near the geodetic mark. Since the multipath is an error source in GPS signals and it refers to the signal reflected from objects in vicinity of a receiver antenna and metal materials around the GPS antenna causes the total reflection to the GPS signal [24], the steel hangar increases the multipath affecting the data. In other experiments, Figure 3b, far from this kind of interference, the mean error did not reach this value. 

The results obtained using this setup show that the standard deviation of the measurements is less than 1 m, which is less than the acceptable uncertainty for this experiment, which is 9.6 m, as shown above. The statistical characteristics obtained in the experiment are shown in Table 1. Therefore, it can be seen that the use of the commercial APS used in this study is feasible for application in VOR inspection.

#### 3.1.2. Dynamic Experiment (Relation among APS Parameters and RPA Flight Mission Parameters)

In this experiment, it was verified whether the APS was able to meet the ICAO requirements for inspecting the VOR system. It was checked whether the distance traveled by the RPA between two data samples were smaller than the maximum error permitted by ICAO regulations. Otherwise, the information lost between two data samples would hide the VOR error that must be detected by the inspection. 

Using the information provided by a DGPS (Differential Global Positioning System) system as reference, it was also possible to analyze how the error in the position information was affected by the APS flying speed, or by the sample rate used in the positioning data acquisition. This experiment analyzes the errors a mission, such as the one proposed in example in Section 3.1, would have. The mission parameters were cruise velocity of 10.5 m/s, APS sample rate 7 Hz, experiment accuracy fixed at 0.1°. This results in a minimum mission radius of 0.4641 NM (859.51 m). 

The DGPS used in this experiment, a DL-V3 NovAtel GNSS receiver, has a maximum acquisition rate of 20 Hz and has an accuracy of 0.45 m [25]. The position in the APS was calculated with a m8n Neo-M8N Ublox GPS receiver, with an acquisition rate fixed at 5 Hz. To increase the actualization information rate and to maintain the operation in case of GPS signal loss for a brief time interval, the AP uses the INS and GPS in its APS. In this way, the APS used has a position accuracy inferior to 2.5 m and 10 Hz maximum acquisition rate [26]. Table 2 presents the characteristics of the GPS and DGPS systems used in these experiments. The DGPS system, with better quality characteristics, was used as the reference signal in the dynamic experiment. However, it is important to note that the results presented were obtained using the APS (GPS and INS working together) operating with a 7 Hz acquisition rate, as proposed in example in Section 3.1.

Two trajectories were implemented to analyze the position information sent by the APS. Trajectory T1 was performed with an average speed *Vm* = 1 m/s and trajectory T2 with *Vm* = 3 m/s. From each trajectory two data intervals were extracted: I-1.1, I-1.2 (trajectory T1), and I-2.1 I-2.2 (trajectory T2). 

Figure 4 depicts the data points acquired in the dynamic experiment, interval I-2.2. There are some missing points in the APS data sequence. This indicates the need to perform more than one lap in the circular trajectory during the VOR inspection mission. When several laps are done, the impact of missing points in each VOR radial is minimized in the mean and standard error computation. 

As stated before, ICAO regulation accepts a maximum uncertainty of 0.6° in the VOR alignment. Considering a trajectory with 0.5 NM (926 m) radius, a 0.6° uncertainty corresponds to 9.6 m. Therefore, to meet the ICAO regulations, the positioning system to be used in VOR inspection must provide position data with an error lower than 9.6 m. In this way, to study the total error, it means that the sum of the standard deviation of the static experiment and the mean error resulting from the dynamic experiment should be lower or equal 9.6 m.

The data from dynamic experiment were sampled with the APS flying at speeds of *Vm* = 1 m/s and *Vm* = 3 m/s (trajectories T1 and T2). As the velocity to perform the VOR inspection using an RPA was defined to *Vm* = 10.5 m/s, the data from dynamic experiment were extrapolated to estimate the mean dynamic error at *Vm* = 10.5 m/s, as follows. 

In our experiment, we had four intervals, I-1.1, I-1.2, I-2.1 and I-2.2, with mean velocities of 1.15 m/s, 1.10 m/s, 2.76 m/s and 3.52 m/s, respectively. These mean velocities were computed by the provided DGPS velocities during the trajectory. Considering *δx* as the distance traveled between two samples, we need to distinguish *δx_exp_*, the distance calculated using the experimental data for each interval, and *δx_10.5_*, which is the estimated distance with the planned VOR inspection velocity, *V_10.5e_* = *V_VOR_mission_* = 10.5 m/s. Using this information, we can fill in Table 3. In each line, we have the considered interval, the mean distance traveled between two samples and the mean velocity used to travel in that interval. The last two columns are related to the extrapolation: the fourth column is the estimated distance that should be traveled between two samples if the travel velocity were the planned VOR inspection velocity, *V_10.5e_* = *V_VOR_mission_* = 10.5 m/s.

According to Table 3, it was possible to determine the estimated *δx_mean_* of the planned mission velocity *V_VOR_mission_* (mean of the ***δx_10.5_*** values), and its standard deviation to the APS, as presented in Table 4. Also, using the same approach to analyze the DGPS information, the mean distance between samples was estimated as 1.06 m, with a standard deviation of 0.01 m. This implies a better quality of information related to the APS. In this way, it is possible to use the DGPS as the reference system in these experiments. 

The positioning system to be used in VOR inspection must provide position data with error lower than 9.6 m, according to the ICAO regulations. The experimental results show that, for the APS used, the distance traveled between two consecutive samples with *V_VOR_* = 10.5 m/s is *δx* = 1.37 m. In this way, the elapsed distance between two samples will be 14% of the maximum ICAO tolerated position error range. For the DGPS system, it is *δx* = 1.06, which is 11% of the ICAO rule. In this way, the experimental data indicates that both APS and DGPS systems are compliant with ICAO regulations with respect to the data acquired in the time interval of a VOR radial. 

#### 3.1.3. Composing the Total Error

To analyze whether the APS is adequate to VOR inspection, the total error, consisting of the static and dynamic error, must be considered. Using the experimental values of the APS (maximum values), the dynamic error will be associated with the maximum positioning error among two sample times. Given the distance traveled among two sample times, in our case *δx_V_VOR_* = 1.53 m (Table 3), and considering the maximum dynamic position error as half this distance, we are very conservative since we are creating error regions with the diameter equal the distance traveled among two sample times. The maximum static mean error is the value found with the static experiment, which in our case was 3.50 m (Table 1). Then, to calculate the total error considering both static and dynamic error, the following expression is proposed:(1)$$error=tendecy+\partial_{V_{VOR}}/2$$

By Equation (1), the instrumental error is computed as 4.27 m. This value corresponds to 0.26° misalignment error in a VOR signal reading using a circular inspection trajectory with 0.5 NM radius. This value, 0.26°, is lower than the maximum error acceptable by the regulation [3], which is 0.3° for radial structures (due to modulation errors) and 0.6° for misalignment errors in the VOR equipment. In this way, even if a calibration routine is not performed before the mission initialization, the results indicate that the APS provides information within standard regulation.

### 3.2. Correction System

On-site self-calibration is a useful procedure for dealing systematic on-board sensor error [27]. Considering the errors present in the measurements, a correction procedure was developed in order to estimate the position error at the beginning of the mission. It is assumed that a geodetic mark is available near the airport location. This correction procedure allows more accurate position information to be provided by the low-cost commercial APS. The procedure for performing the initial calibration can be described as follows:The system is placed at a geodetic mark at the airport. Since the RPA platform is a medium-size aircraft, the placement of the RPA on the mark should be able to be performed without major difficulties.The system takes several samples during a pre-determined time interval and uses them to compute the mean values of latitude and longitude. In this work, the time interval was 5 min.The error between the position of the known geographic mark and the estimated position value are computed. This error value is used as the correction factor during the mission.

The values of latitude and longitude, adjusted by the correction factor, need to be encapsulated in a standard GPS package format in consideration of the checksum parameters. This is done by the Correction System. It is important note that, as the Earth is not a plane and is not even a perfect sphere, the computed error is valid at that point. At another point on the globe, this error distance, which is actually an arc, will not correspond to the same error distance. However, as we are proposing a system for carrying out VOR inspection, this calibration procedure uses marks in the airport at which the VOR is located, and the flight distances are small, it is possible to consider the mission to be taking place in a plane tangent system. This method of computing the error between the position of the known geographic mark and the estimated GPS position value is also used in [28,29].

An experimental test to verify the operation of the Correction System in the APS data was performed. Two data sets using the geodetic mark 2 (see Section 4.1) were taken. Dataset-1 was taken forcing a 20 m error and dataset-2 a 25 m error. After 5 min, the correction system calculates the position error and uses this information to correct the position readings. Once the correction error factor has been determined, the system uses it to correct the new position readings.

Table 5 lists the mean error and standard deviation of the two data sets (dataset-1, dataset-2) for a period of 10 min, arbitrarily chosen with the aim of providing appropriate statistical information. Comparing these results with Table 1, it is possible to verify that the mean error decreased by 91.1% for latitude and 61.8% for longitude, for dataset-1. The standard deviation values changed; however, this may not necessarily be attributable to the correction used by the correction system addition. This is because the GPS signal received depends on additional factors, including satellite geometry, signal blockage, and atmospheric conditions [30]; factors that were not monitored during our experiment. In this correction system, the correction factor is calculated as an average of the error between the known geographic coordinate known and the measurements given by the GPS, and its associated standard deviation. This average correction factor is estimated at the beginning of the mission and used during the mission. A real-time correction system in which the factor would be estimated along the mission could be an improvement of the correction system. These results show that the correction system proposed improves the position information for use in the VOR inspection. 

## 4. The HIL Platform and Mission Parameters

To perform a mission in the HIL platform, it is necessary to determine the parameters of the mission, because they affect the quality of the flight inspection. Also, some elements must be defined in order to be used in the automatic flight inspection procedure. The development required for a HIL platform to work appropriately for the task in question is presented in this section.

Another problem to be addressed in the VOR inspection using RPA is the definition of the inspection trajectory to be followed. To automatically inspect a VOR station, it is the system should compute the set of WPs necessary for the aircraft to complete the flight mission. Then, with information about the quality of the positioning system and the automatic trajectory definition implemented, a HIL platform can be used to study the validity of applying RPAs for VOR inspection.

### 4.1. HIL Platform

Figure 5 depicts the diagram of the HIL platform developed, presenting the subsystem executed in the Personal Computer (PC), the subsystem AP (the hardware of the AP used), and the signal information traveling between them. The PC executes the flight simulator, which provides the attitude and position of the aircraft to the Mission Planner. The Mission Planner converts the data between the simulator X-Plane [31] and the AP. The XPlane flight simulator has FAA certification and is widely used due its communication characteristics and its realistic flight model, which is based on blade element theory [32,33]. The AP hardware executes the control mission algorithm using the pre-registered WPs as input data to determine a trajectory for the VOR inspection. With the receipt of the aircraft’s attitude and position information from the flight simulator, and the registered WP information, the algorithm calculates the surface deflection in order to track the planned trajectory. The computed deflections are sent back to the flight simulator, closing the HIL loop. The Data Logger records the aircraft position data and the VOR signals acquired during the mission for posterior analysis. The Processing block is discussed in Section 5. The AP sensors are not used in the HIL platform, and the signals necessary for closing the control loop are received from the simulator and are considered to be ideal.

### 4.2. Routine for Automatic Generation Inspection Trajectory—Mission Automation

The trajectory has three phases: taking off over the airport runway; approximation of the inspection trajectory; and inspection trajectory. The complete inspection mission is presented in Figure 6. 

The developed MATLAB^®^ routine for generating a VOR inspection trajectory automatically generates all the WPs. The AP system allows this set of WPs to be registered in the mission planner before the mission and to create a flight mission data log for subsequent analysis. Both registers are produced by means of text files. The routine must consider some aspects of the mission: the RPA system characteristics (cruise velocities, positioning system sample rate), and the airport in which the VOR to be inspected is located (the runway airport heading). In this way, the routine generates the trajectory in consideration of the aircraft parameters and the mission data as the input. The output is a set of WPs to be followed by the aircraft mission along the three phases of the inspection.

The first phase of the mission, taking off from the airport runway, is accomplished by following six waypoints with increasing altitudes. The approximation, the second phase of the mission, is an arc connecting the takeoff straight trajectory to the mission inspection trajectory. The arc is composed of thirty WPs.

To calculate the radius of the inspection trajectory, the third phase of the mission, several parameters must be considered. The aircraft positioning system (embedded in the AP hardware) should be able to map each sample of the VOR reading inside a region, ensuring the required accuracy of 0.6° to the VOR alignment verification [3]. The cruise velocity in which the mission will be performed, and the sample rate used to VOR signal reading must be established. In this way, the minimum circle radius of the mission inspection can be computed by:(2)$$R=\frac{vel/rate}{2\pi .∆r}\times 360$$
where *vel* is the cruise speed (m/s), *rate* is the positioning system sample rate (Hz), Δ*r* is the required accuracy, and *R* is the trajectory radius (m), the distance between the considered VOR and the airplane. The waypoints are equidistant from one another.

As an example, consider a cruise velocity of 10.5 m/s, an APS sample rate of 7 Hz, and an accuracy of 0.1° (lower than the one required by regulation). Thus, the minimum radius to execute the mission is 0.4641 NM (859.51 m). Considering these data, two trajectories were automatically generated for the inspection of the VOR in SJK and VCP airports, as illustrated in Figure 7. 

These airports were chosen because they are near the research center where this study was conducted. Another reason is that they have different characteristics with respect to the automatic generation of the inspection trajectory. At SJK airport, the automatic trajectory is such that the approximation phase trajectory is outside the inspection trajectory. At VCP airport, the entrance is inside, as can be seen in Figure 7a,b, respectively. In both cases, the takeoff starts at runway 33 and proceeds to an inspection trajectory of 0.5 NM (926 m) radius at 90 m of height. The approximation procedures, however, are not the same, due to the different VOR locations in relation to the runways. 

To accomplish the VOR inspection, the aircraft must execute a circular trajectory crossing all radials at least once, and some considerations regarding the use of RPA, such as those being proposed, must be made. The use of a medium-size RPA makes it possible to minimize the trajectory radius and the use of a less accurate positioning system. Thereby, it is possible to make more than one VOR signal sample acquisition in each radial, and it is also possible to proceed more than one lap during the mission. In this way, it is possible to determine the statistical error modeling for each radial and for the complete system.

### 4.3. Radius Trajectory and Its Influence on Misalignment Error Estimation

To determine the misalignment error, an acquisition and data analysis procedure was developed. According to the standard, the misalignment error is the mean error of the 360 radials [3]. Figure 8 shows regions and elements of interest around the VOR: *Radius*, *Width*, *Center*, *Limit*, *Decision Region*. These regions and elements are parameters used in the algorithm responsible for comparing the position information given by the VOR system and the position information given in the APS during the mission. When the RPA crosses the decision region of each radial, the algorithm must detect misalignment errors. It is in this region that VOR misalignments can be detected.

The *Radius* (*R*) is a parameter that is to be defined before the mission takes place. The *Width* (*W*) is the radial width and is a function of the radius of the trajectory given by:(3)$$W=2\pi R/360^{{}^{\circ}}$$

The *Center* is the radial from the origin to the VOR station, and its azimuth is the intermediate value between two adjacent radials. The *Limit* is a line starting from the origin, and its azimuth is an integer number (in degrees). The *Decision Region* (*Rgn*) is the region where the data used in the algorithm is sampled. It occurs around the limit line, beginning Δ*r* before the limit line and finishing Δ*r* after it:(4)Rgn=Limit±∆r
with Δ*r* = 0.2° (this value assures that at least one sample is taken by our specific SPPA within the uncertainty region acceptable by regulation). This region was conceived to consider the APS acquisition rate used in this work. All VOR data are read; however, they will be stored only if the plane is within the decision region.

To calculate the mean value of the radial, a routine determines the points that should be stored, i.e., the points inside *Rgn*. The mean value of each radial is found by computing the mean among the positions of all points stored in each region *Rgn*. The mean and standard deviation of each of the 360 data sets acquired are computed and compared with reference values. 

The value of the VOR radial modeled as a function of the receptor and the VOR station position, *θ_modeled_*, is given by:(5)$$\theta_{modeled}=360^{{}^{\circ}}-\arctan((lat_{recep}-lon_{VOR})/(lat_{recep}-lon_{recep}))-\delta_{mag}$$
where *lat_VOR_* is the latitude information given by the VOR system, *lon_VORr_* is the longitude information given by the VOR system, *lat_recep_* is the latitude information given by the APS, *lon_recep_* is the longitude information given by the APS, and *δ_mag_* is the magnetic declination, available in [34].

The parameter *θ_modeled_* is adjusted to avoid a numerical jump between radial 359° and 0°. Values greater than 359.5° are considered to be in the decision region of radial 0°. In this way, the *θ_modeled_* assumes values in the range of −0.5° < *θ_modeled_* ≤ 359.5° and can be compared with *θ_read_*, the information given by the APS. Because of this, it is possible to use classical statistical analysis, instead of circular statistics, with the VOR’s circular variables.

Due to the sample rate of the positioning system and the width of the decision region, the number of samples of each radial depends on the trajectory executed by the airplane. Due to this, missions with different radius trajectories were carried out to study the error in VOR signal readings. The error in VOR signal, in the context of VOR flight inspection, is computed by comparing the information given by the VOR equipment (the radial information in which the airplane is, given by the VOR, at each moment, received by the airplane) and the airplane position given by the positioning system used for the flight inspection (in our case, the APS). In all missions, the number of WP was 120, and the airplane flew in two complete circles around the VOR station. Table 6 shows the misalignment errors (*ϵ = θ_modeled_* − *θ_read_*) in each mission. *ϵ_max_* corresponds to the largest error among all 360 radials, and the *ϵ_mean_* corresponds to the VOR station misalignment error, computed by the mean error of the 360 radials.

### 4.4. Number of Flights around VOR Station and the Influence on Misalignment Error Estimation

Considering the minimum radius determined in Section 4.1, *Rmin* = 859.51 m (0.46 NM), and the misalignment errors found in the HIL mission, another mission with a 926 m (0.50 NM) radius was executed. This new value is reasonable, as there is little to no variation in the error if the radius is increased. In addition, the number of flights around the VOR station is supposed to change the misalignment error computed. In this way, the misalignment error was also studied as a function of the number of complete circles around the VOR station executed by the airplane, and the results are presented in Table 7.

The results show that when increasing the number of flights around the station, the misalignment error decreases. However, there is a compromise between the number of turns and the time of autonomy of the RPA. In addition, since no significant reduction in error occurs for more than three flights, they were limited to three.

## 5. Results of HIL Simulations for VOR Inspection

This section presents the VOR inspection tests executed using the developed HIL test platform. The parameters used in the flight mission, determined in the previous sections, are summarized in Table 8. Height equal 90 m corresponds to an elevation angle of 5.5° to the chosen mission radius, which is within the limits, between 4° and 6°, established by regulation [23]. 

The HIL platform diagram was presented in Figure 5. Two types of errors were added in the processing dataflow block in order to verify whether the HIL platform could detect when a VOR station is not working within the norm specification, see Figure 9. The first type is an error inserted in the positioning information, due to errors in latitude and longitude position, *ϵ_Lat* and *ϵ_Lon*, respectively. One constant and a random portion generated *ϵ_Lat* and *ϵ_Lon*. In Table 9 they are presented as (mean ± standard deviation). The second type, *ϵ_VOR*, corresponds to VOR’s reading errors (station misalignment and modulation error). There are two components encapsulated in *ϵ_VOR*: *vor*, a constant value, which shifts the value of the radial read from the Data Logger; and a modulation error, as specified by ICAO. The *ϵ_VOR* is given by the equation:(6)ε_VOR=vor+(10.5×sin(36.25×vor)+1.5

For each case of interest, VOR misalignment or VOR modulation error portion is active, and Table 9 presents the errors added in each of the five tests performed using the HIL platform.

To determine the error in the radials, *ϵ_radials*, the system computes the difference between the reference value and the mean of the readings during the mission. This generates an error value in each radial. The mean of the error of these 360 radials is computed to determine the error of the station under inspection. In addition, it is also determined the radial at which the largest error occurs and the misalignment error of the VOR station.

Figure 10, Figure 11 and Figure 12 depicts the data acquired during the flight inspection using the HIL platform. The coordinates *X* and *Y* in these figures represent the coordinate frame centered in the VOR station. Figure 10a presents a complete circular trajectory around the VOR equipment. From Figure 10b to Figure 12b it is presented the region in the flight trajectory in which the largest error occurred. In each figure, it is presented the radial centers along the trajectory traveled by the RPA and the position information sampled using the APS, for the five tests listed in Table 9. In this way, by these figures, it is possible to see the impact of the different types of errors added in the signal.

The first test, *Test 1*, was a reference mission and no errors were added to it. Figure 10a shows the circular trajectory executed by the RPA and Figure 10b shows the region with the largest identified errors in this trajectory. It is possible to observe that the sample points corresponding to the radials are grouped and aligned, since there is no error added.

In *Test 2*, positioning errors were added with characteristics as the identified in the commercial AP studied in Section 5, as listed in Table 9. Figure 11a shows the Radial with largest error. It can be noted the dispersion in the sampled points, due to the random error added in the system. In this Radial it was detected a misalignment of 0.02 ± 0.00°, resulting in a VOR aligned according to ICAO’s regulation.

In *Test 3*, a 4° misalignment error in the VOR station signal was included by adding this misalignment in each VOR reading, as listed in Table 9. Figure 11b presents the Radial in which the largest error was identified in this test. It is possible to note the dispersion in the positioning points as in *Test 2*. However, the group of points corresponding to a radial is displaced by four radials. In this Radial it was detected a misalignment of 3.99 ± 0.00°. This implies a misalignment greater than 2.00° at the VOR station, which is the maximum value allowed by the regulation. In this way, the result given by the HIL test platform is that this VOR station is misaligned and it should not be approved by the inspection.

In *test 4*, positioning and modulation errors were inserted in the VOR station, as listed in Table 9. Figure 12a shows that the signal corresponding to the Radial 104 is received in a spread region due the inserted modulation error. In this Radial it was detected a misalignment of 0.17 ± 0.04°, resulting in a VOR aligned.

*Test 5* used the same errors of *test 4* and included and additional 4° misalignment error, as listed in Table 9. The result obtained is similar to *test 4*, differing by the readings rotated by 4°, which corresponds to the misalignment added, see Figure 12b. This results in a VOR misaligned.

A summary of the results given by the HIL test platform is listed in Table 10. The results show that the APS used can provide the position information to the VOR inspection identifying the cases in which the VOR station was misaligned. In other cases, even though having modulation errors inserted in some radials (*tests 4* and *5*), the system was able to verify the VOR alignment situation.

## 6. Discussion

NavAids provide electronic guidance for aircraft, helping navigation in poor visual conditions. According to regulations, the NavAids must pass periodical inspections to ensure calibration and air system safety. These flight inspections require special air traffic operations and are a repetitive task that has to be performed in each airport with NavAids. The inspection of VOR requires the displacement of a well-equipped aircraft to the corresponding airport, and, as these inspections require special air traffic operations and include repetitive tasks, the use of RPA in VOR flight inspection is an alternative solution.

ICAO regulations establish the maximum uncertainty acceptable in the VOR alignment inspection and the conditions under which such inspections must be taken. This work assumes that the RPA is equipped with a commercial AutoPilot that can perform the flight mission automatically. Initially, experiments to verify the errors associated with the positioning system of the AP used were conducted. Two tests used geodetic marks in SJK airport as reference landmarks. In one test, the mark was close to a metallic structure, resulting in a higher error, and the other test resulted in a greater dispersion due to unstable weather conditions. However, in both tests, the errors were within the limits permitted by the regulations. 

Since the commercial AP used provides position information with an accuracy deemed acceptable by the regulations, the AP was tested in a HIL platform. To develop automatic missions in the HIL platform, the influence of some parameter variations on misalignment error detection was studied. In addition, a Decision Region limits the data sampling by the algorithm. This region was defined to reduce the stored data within the region of interest.

Finally, five mission flight inspections were performed using the HIL test platform. The first mission was a reference mission, with no introduced error in the VOR signal received from the flight simulator used. In *Test 2*, a random signal was added into the VOR signal, simulating errors in the positioning system. These errors correspond to the dynamic and static errors found experimentally using the positioning system of the commercial AP used. In *Test 3*, in addition to the random signal in the second test, error was also introduced into the VOR signal, simulating a misalignment in the equipment. In *Test 4*, the VOR misalignment was not present, instead, a VOR modulation error was introduced. Finally, in *Test 5*, all errors were introduced (positioning, VOR alignment and VOR modulation). The results given by the automatic VOR inspection in *Test 3* and *Test 5* indicated that the VOR presented a misalignment.

The use of RPA for flight inspections involves many aspects, as shown in the works available in the literature. References [10] and [11] do not evaluate the proposed architecture by means of HIL. Using equipment specifications, a flight simulator, and an autopilot, it was concluded that a small or medium RPA is capable of performing a flight test inspection. Different noise function parameters were tested, and an alarm signal was generated during the simulations when the error reached the maximum established by the regulations. Unfortunately, no numerical data regarding the noise level was given for comparison. 

The authors of [12] supported their analysis employing consistent experiments using a tailor-made UAV capable of embarking a certified flight analyzer. As already stated, no test regarding the measurements of a VOR was performed, but the results regarding the accuracy of the UAV trajectory indicate the feasibility of using UAs for VOR flight inspections. 

The UA used in [13] had the largest payload capacity (50 kg, 50 cm^3^) and the best flight autonomy (10 h) among the works available in the literature. The authors reported successful taxiing maneuvers at the airport and a reliable data link, but flight tests were not performed. Considering the continuous miniaturization of Flight Inspection Systems (FIS), the UAV used in [10] will soon compete with small size flight inspection aircrafts. 

References [14,15,18] presented their results in measuring flight guidance parameters in the time and frequency domains. The aim of the developed platform and experiments was to measure the influence of wind turbines on terrestrial navigation signals by analyzing the integrity of the RF signal. This was research with a different scope from our work, as we consider RF signals without any integrity problems. This research demonstrated the use of UAs as a valuable tool supplementing conventional flight inspections in navigation aids.

References [5,10] pointed out the problem of defining the flight inspection trajectory using waypoints. Instead of using a routine for the generation inspection trajectory, as described in Section 4.2, Reference [5] proposed a flight plane based on “legs”. Four different “legs” were defined, which can be understood as being like commands in a code; e.g., an “interactive leg” can be used for repeating flight sequences. This proposal helps in generating an inspection trajectory, but still requires considerable interaction from the person in charge of defining the trajectory. The routine for automatic generation of inspection trajectory, described in Section 4.2, works at a high level with less interaction of the person in charge of defining the trajectory. The inputs are the RPA system characteristics, and the airport at which the VOR to be inspected is located. On the other hand, as a flight plan design environment, using an XML-based specification, the proposal in [5] allows the description of any flight inspection trajectory, i.e., it is not limited to a VOR flight inspection. As with other proposals, Reference [12] uses an UAV that follows waypoints in a programmed path for the measurements. No comments regarding a tool for helping the definition of flight inspection trajectories is given.

## 7. Conclusions

The APS errors were determined by tests, and errors in latitude and longitude of (1.65 × 10^−06^, 4.91 × 10^−06^) and (5.08 × 10^−06^, 7.01 × 10^−06^) (mean, standard deviation) were used in the HILs tests. The misalignment observed due to these errors was 0.02 ± 0.00°, a value below the limit defined by the ICAO regulations.

The HIL platform allows the insertion of misalignment and modulation error into the VOR station. The HIL Processing block is responsible for this task. The introduction of a 4° misalignment error in the VOR station results in a 3.99° misalignment during the flight test, a value greater than the maximum one allowed by the regulation, 2.00°. When introducing a 4° misalignment error into the VOR station with modulation error, a misalignment of 4.02° was observed, thus also resulting in successful misaligned detection.

Due to the extensive number of tests, the Routine for Automatic Generation Inspection Trajectory proved to be a useful tool. Without this tool, the generation of the way points for complete flight inspection would be tedious and prone to error.

All test results point to the feasibility of using Remotely Piloted Aircrafts in VOR flight inspection. However, other experimental tests related to the accuracy of the positioning information of the AP system and the performance in different weather conditions should be analyzed. Wind and other perturbations along the VOR flight inspections can cause the aircraft to deviate from its pre-defined trajectory [35]. However, the most important information for the flight inspection, the comparison between the position (heading) informed by the VOR equipment and the position informed by the APS, is not affected by the wind. One has also to take into account that the costs involved in a flight test using Remotely Piloted Aircraft are much lower than those for tests using a certified aircraft. In this respect, the wait for appropriate weather conditions will impose a minimal cost increase.

## Figures and Tables

**Figure 1 sensors-20-01947-f001:**
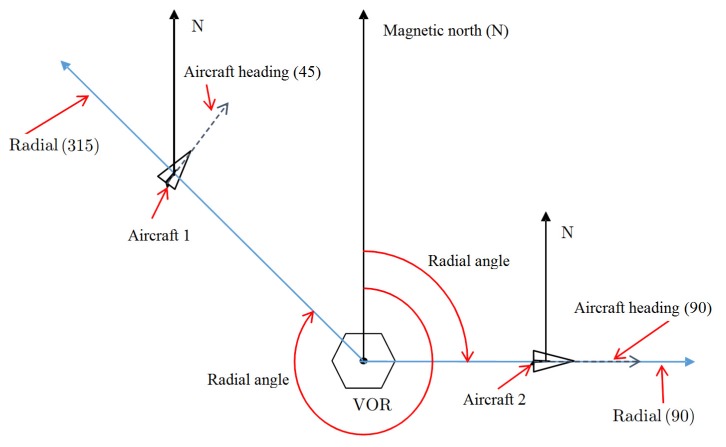
VOR radials and trajectory information obtained by planes receiving VOR signal.

**Figure 2 sensors-20-01947-f002:**
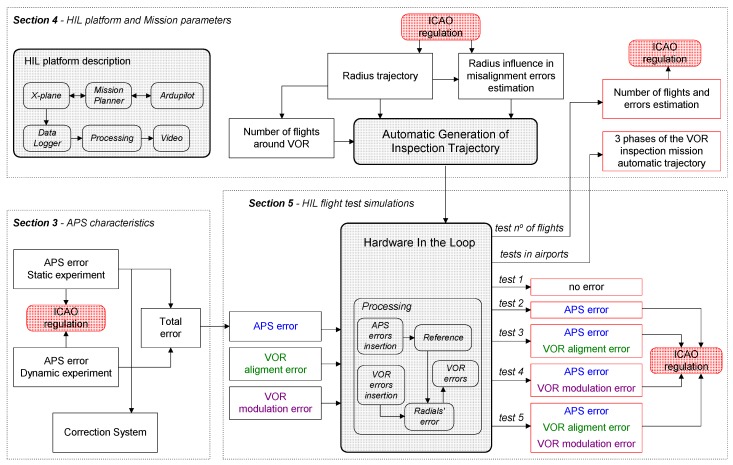
Flowchart showing the global approach for assessing the feasibility.

**Figure 3 sensors-20-01947-f003:**
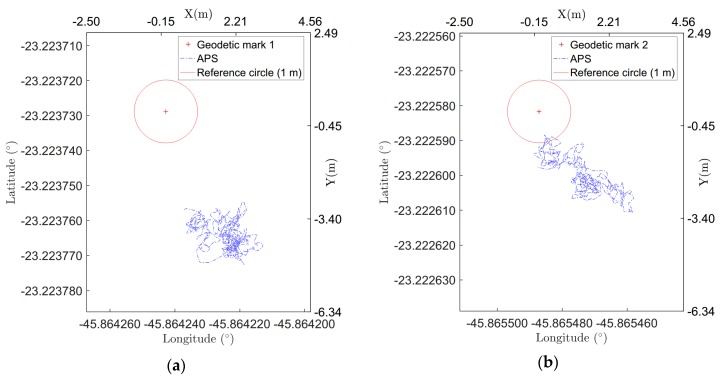
Geodetic mark (+) inside of a one-meter circle, and the APS position samples: (**a**) geodetic mark 1, (**b**) geodetic mark 2.

**Figure 4 sensors-20-01947-f004:**
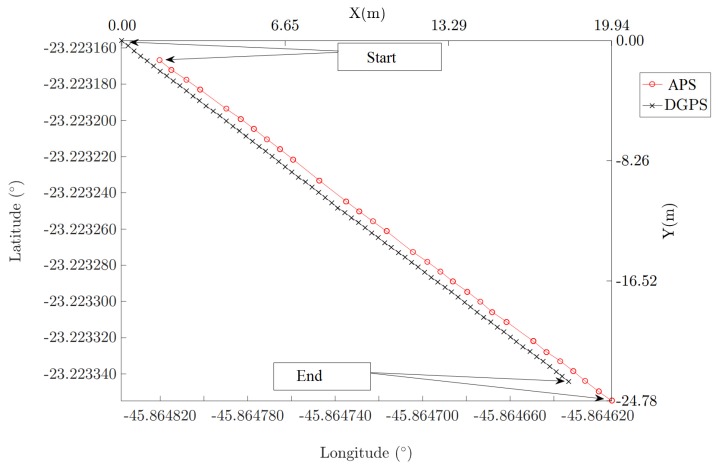
Data points acquired in the dynamic experiment, interval I-2.2.

**Figure 5 sensors-20-01947-f005:**
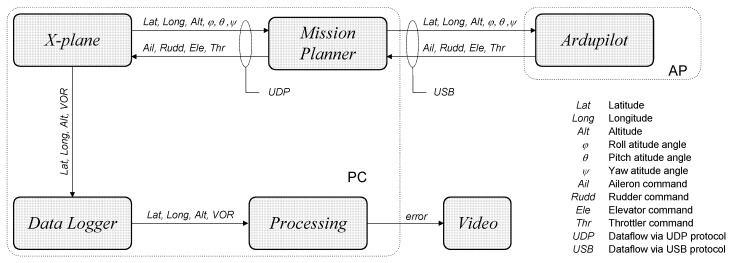
Developed HIL platform. An expansion of the block *Processing* is depicted in Figure 9.

**Figure 6 sensors-20-01947-f006:**
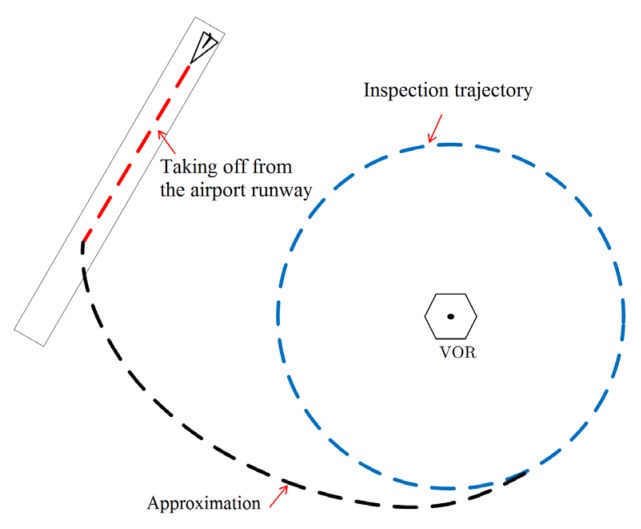
The three phases of the VOR inspection mission automatic trajectory.

**Figure 7 sensors-20-01947-f007:**
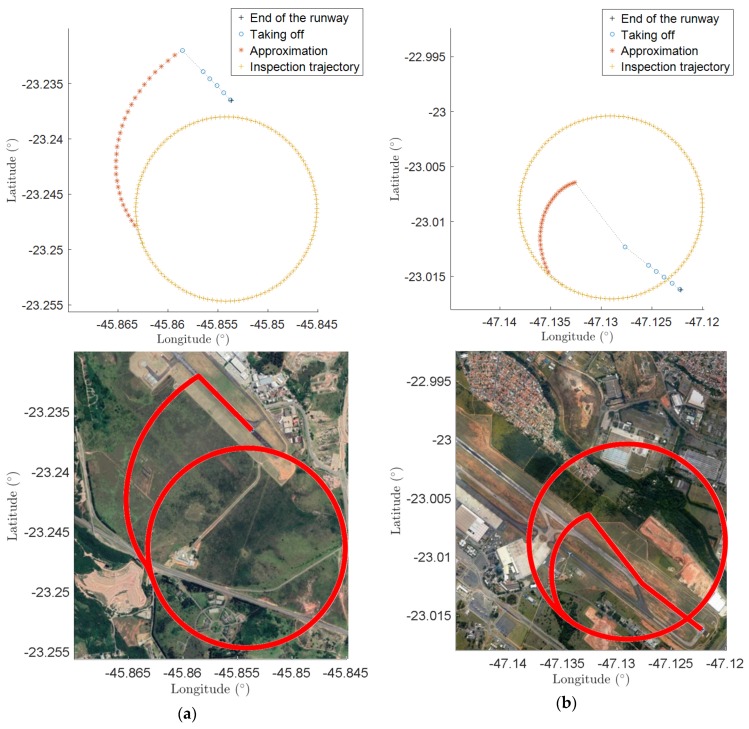
Automatically generated trajectories. Longitude and latitude and its drawing using Google Maps: (**a**) SJK (São José dos Campos—Brazil) airport; (**b**) VCP (Campinas—Brazil) airport.

**Figure 8 sensors-20-01947-f008:**
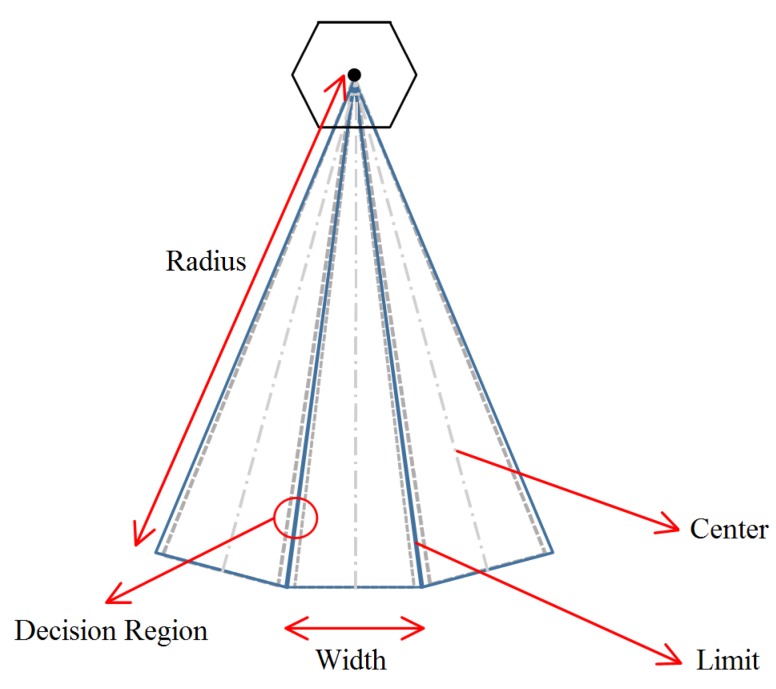
Regions and elements of interest for VOR misalignment analysis.

**Figure 9 sensors-20-01947-f009:**
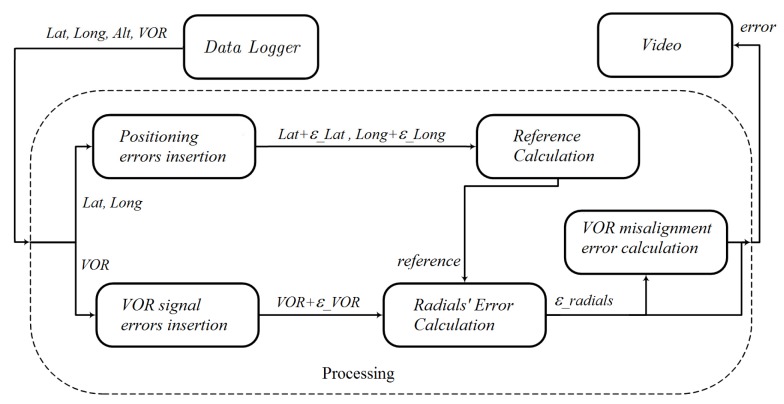
Mission dataflow. The *Processing* block is part of the complete HIL platform depicted in Figure 5.

**Figure 10 sensors-20-01947-f010:**
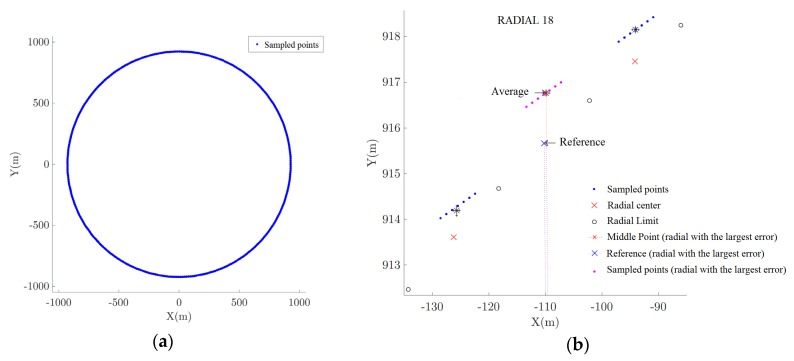
Mission flight using the HIL platform: (**a**) Inspection trajectory in *Test 1*; (**b**) Radial with the largest error in *Test 1*.

**Figure 11 sensors-20-01947-f011:**
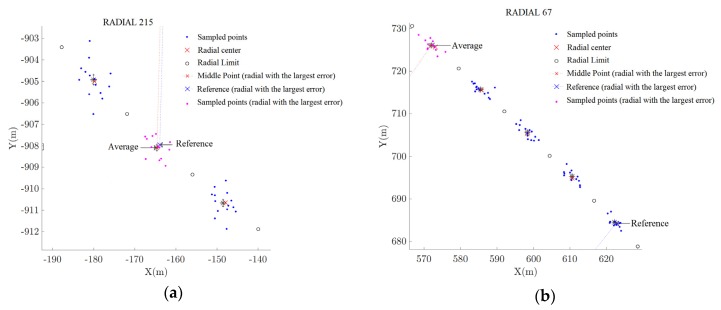
VOR inspection using the developed HIL platform: (**a**) *Test 2*; (**b**) *Test 3*.

**Figure 12 sensors-20-01947-f012:**
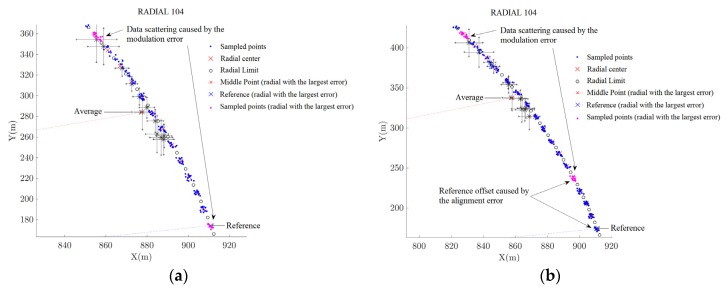
VOR inspection using the developed HIL platform: (**a**) *Test 4*; (**b**) *Test 5*.

**Table 1 sensors-20-01947-t001:** Mean error, *ϵ_mean_*, and standard deviation, *σ_m_*, in comparison with two geodetic marks.

Geodetic. Mark	Latitude				Longitude			
	*ϵ_mean_* (‘’)	*ϵ_mean_* (m)	*σ_m_* (‘’)	*σ_m_* (m)	*ϵ_mean_* (‘’)	*ϵ_mean_* (m)	*σ_m_* (‘’)	*ϵ_mean_* (m)
1	1.30 × 10^−1^	3.9807	8.14 × 10^−5^	0.0025	−6.88 × 10^−2^	2.112	1.09 × 10^−4^	0.0034
2	6.66 × 10^−2^	2.0594	1.14 × 10^−4^	0.0035	−4.79 × 10^−2^	1.4805	1.63 × 10^−4^	0.0051

**Table 2 sensors-20-01947-t002:** Characteristic of the GPS and DGPS systems used.

Equipment	Accuracy (m)	Precision (m)	Maximum Acquisition Rate (Hz)	Used Acquisition Rate (Hz)
DL-V3 (DGPS)	0.45	<0.25	20	10
m8n (GPS)	2.5	-	5	5

**Table 3 sensors-20-01947-t003:** *δx_10.5_* estimated for *V_10.5_* = 10.5 m/s by extrapolating the experimental data *δx_exp_* and *V_exp_*.

Interval	*δx_exp_* (m)	*V_exp_* (m/s)	*δx_10.5_* (m)	*V_10.5_* (m/s)
I-1.1	0.15	1.15	1.37	10.5
I-1.2	0.16	1.10	1.53	10.5
I-2.1	0.33	2.76	1.26	10.5
I-2.2	0.44	3.52	1.31	10.5

**Table 4 sensors-20-01947-t004:** Estimated *δx_mean_* to *V_VOR_*, and its standard deviation.

*V_VOR_* (m/s)	*δx_mean_* (m)	*σ(δx)* (m)
10.5	1.37	0.05

**Table 5 sensors-20-01947-t005:** Mean errors and standard deviation with the developed correction system application.

Dataset	Latitude				Longitude			
	*ϵ_mean_* (‘’)	*σ_m_* (‘’)	*ϵ_mean_* (m)	*σ_m_* (m)	*ϵ_mean_* (‘’)	*σ_m_* (‘’)	*ϵ_mean_* (m)	*σ_m_* (m)
1	5.94 × 10^−3^	2.07 × 10^−4^	0.1824	0.0064	−1.82 × 10^−2^	1.98 × 10^−4^	−0.5655	0.0061
2	1.27 × 10^−2^	2.16 × 10^−4^	0.3903	0.0066	−2.69 × 10^−4^	2.95 × 10^−4^	−0.0083	0.0091

**Table 6 sensors-20-01947-t006:** Misalignment error as a function of the trajectory radius.

Trajectory Radius (NM)	Trajectory Radius (m)	*ϵ_max_* (‘’)	*ϵ_mean_* (‘’)
0.25	463	259.20	72.00
0.50	926	110.80	36.00
0.75	1389	108.00	36.00
1.00	1852	100.80	36.00
1.25	2315	115.20	36.00
1.50	2778	108.00	36.00

**Table 7 sensors-20-01947-t007:** Misalignment error as a function of number of flights around the VOR station.

Number of Flights	*ϵ_max_* (‘’)	*ϵ_mean_* (‘’)
1	252.0	72.0
2	216.0	72.0
3	180.0	32.4
4	129.6	32.4
5	129.6	32.4
6	129.6	32.4

**Table 8 sensors-20-01947-t008:** Misalignment error as a function of the trajectory radius.

Radius of Trajectory (NM)	Number of Turns Around the Station	Height (m)
0.5	3	90

**Table 9 sensors-20-01947-t009:** Errors inserted in the system for each test.

Test	Positioning APS		VOR Alignment	VOR Modulation
	Latitude (‘’)	Longitude (‘’)	(^o^)	(^o^)
1	no error	no error	no error	no error
2	5.35 × 10^−3^ ± 1.59 × 10^−2^	−1.65 × 10^−2^ ± 2.27 × 10^−2^	no error	no error
3	5.35 × 10^−3^ ± 1.59 × 10^−2^	−1.65 × 10^−2^ ± 2.27 × 10^−2^	4.0	no error
4	5.35 × 10^−3^ ± 1.59 × 10^−2^	−1.65 × 10^−2^ ± 2.27 × 10^−2^	no error	*ϵ_VOR*
5	5.35 × 10^−3^ ± 1.59 × 10^−2^	−1.65 × 10^−2^ ± 2.27 × 10^−2^	4.0	*ϵ_VOR*

**Table 10 sensors-20-01947-t010:** The results given by the develop HIL platform.

Test	Radial with Largest Error			VOR	
	Radial	Radial Mean (^o^)	Error (m)	Misalignment (^o^)	Result
1	18	18.04 ± 0.07	0.49 ± 1.06	0.01 ± 0.00	Aligned
2	215	215.06 ± 0.07	0.99 ± 1.11	0.02 ± 0.00	Aligned
3	67	71.06 ± 0.07	65.54 ± 1.20	3.99 ± 0.00	Misaligned
4	104	111.10 ± 1.10	114.77 ± 17.77	0.17 ± 0.04	Aligned
5	104	114.66 ± 1.08	172.33 ± 17.43	4.02 ± 0.03	Misaligned

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
