# Peer review of "The Feasibility of Remotely Piloted Aircrafts for VOR Flight Inspection"

_sensors, 2020, doi:10.3390/s20071947_

Round 1
Reviewer 1 Report
Dear
I congratulate you for the work presented. I really liked the flowcharts used to demonstrate the methodologies and tests implemented. I pointed out some suggestions in the text asking for improvements in writing and more details of the methodologies and results. The use of very short paragraphs could be revised in some parts of the text.
I pointed out some suggestions in the use of terms commonly used in Cartography and Geodesy (directions and azimuths, for example).
I request special attention to chapter 3. In this chapter I suggested several changes that in my view can enrich the good understanding of the work.
Regarding the tests implemented: would it be possible to justify the choice of the two airports?
Cheers.

Reviewer 2 Report
The manuscript provide information on the feasibility/use of remotely-piloted aircraft for VOR flight inspection.
The scope of the article is proportional to the information value of knowledge contained in it.
The content of the article is supported with elaborate experimental evidences (performing flight tests by implementing a Hardware In the Loop platform).
The introduction contains general information on the solved problem, lacking references or quotations to support such statements. It should contain more information about the state of the current issue, as well as related works, with its quotations and detailed explanations. This part suffers from a complete systematic review of solutions similar to the one proposed by the authors.The authors need to use more contents to stress their research motivation that study on the problem of using remotely-piloted aircraft for VOR flight inspection. At the end of the introduction section, there should be a description on the organization of the paper.
The discussion section should include a much more detailed description of the achieved results, their evaluation as well as the way of verification of the results´ validity.
The conclusion should contain information about a systematic review of other similar proposals that should also be used as a common tool in the evaluation of the achieved results, comparing the results obtained by the authors' research and others existing researches in the field.
Reviewer 3 Report
Original Submission – Recommendation:
Major Revision
Comments to Author:
Dear Ref. No.: Sensors-737637
Title: The feasibility of Remotely-Piloted Aircrafts for VOR flight inspection
0- Abstract:
I think there is no clear goal of what your paper is aiming. I made specific comments on the introduction. I also think that the keywords could extended, not abbreviations.
1- Introduction:
Better discuss [7] and [8] on line 45. Same for [9], [10], [11]. You must link the parameters of system requirements, costs and weight, for all this papers with specific and useful information presented into the introduction, since the paper does not have a related work section. I think that the introduction needs major improvements regarding literature. You should cite more similar works and better explain how these papers are linked to yours.
[Line 55] - It is the first time HIL appears on the intro, write extended name
Last 3 paragraphs show the goals of the paper considering a methodological summary (basically you mention all steps you are going to discuss on the methodology and results), however it would be much clear to identify the scientific aim of the paper within a simple sentence. Where did science advanced with your discovery? Was by the improvement of the hill platform with your code? Was with the AP hardware and the flight simulator? Was by the routine of the inspection trajectory? I would like to see things a bit clearer on these 3 last paragraphs of the introduction.
Section 2. Materials and methods
[Lines 69-117] VOR is well explained here and also has an example of ICAO regulations. This section is fine to me.
Section 2.1. This section is also fine. Figure 2 helps to understand what discussion is expected on each section of the paper.
Section 3. APS
Line 151 and 152, the coordinates have a big blank space between then, should you add something like a semicolon to separate both?
Why information was taken for 40 minutes? Better precision or something like this? You could better explain this on the text.
Figure 3 (b) I think the legend should be geodetic mark 2, right? I would also make both figures with the same size, I realized that x and y scales are different and this is not good to a visual comparison of the results.
I would also shift table 1 right under figure 3, since the information is correlated and it would better improve visualization.
[Lines 166-170] Is there a reference you can add here? Your discussion makes sense, but a reference would make it stronger to the reader.
[Line 182] I think it is the first time that DGPS appears, so write it in extent.
Figure 4 seems a bit blur to me, if it is possible, add a better resolution graph.
[Line 251] You need a reference for equation 2, or you are proposing that it should be calculated this way?
4- HIL platform and mission parameters
[Line 342] It is presented equation 1, but you have previously presented equation 2. I recommend checking the equation numbers for the entire paper.
Follow the same recommendations for Figure 3 to Figure 7. Add same axis scale for both trajectories. I would also like to see a scale bar on the images from google earth, since b) appears to be much closer to the ground than a).
Same problem of blurring for figure 8. I think you can improve quality of all figures by the way (also a lit bit bigger font size on the axis and legend, maybe add legend on the bottom for the figures of section 5).
Section 5 – Results
The results presented are fine and the section is well written.
Section 6 – Discussion
This section is poor. You basically mentioned your results again but you need to compare what you obtained with other similar works (there is not a single reference in this entire section by the way). You mentioned a few on the introduction, which I recommended improving. By better discussing each paper on the introduction this will give you a better base to improve the discussion here. Some questions that you may use to improve the discussion: Who else studied VOR alignment? What trajectory errors were obtained on other studies? Does RPA contribute to these errors? Are different methodologies that led to similar/different results?
I also noticed that you used only 21 references, which is quite a few. I understand that sometimes the research topic is very restricted but you can add literature which is not that specific to compliment your intro and your discussion.
Best regards,
Reviewer.
Round 2
Reviewer 3 Report
Dear Ref. No.: Sensors-737637 (round two)
Title: The feasibility of Remotely-Piloted Aircrafts for VOR flight inspection
There were several improvements on the introduction, better discussing some state of the art papers and also to make paper goals a bit more clear. On the methodology, all my questions were answered and added to the document (including figure suggestions). The section with most improvement was the discussion. As mentioned by the authors, five paragraphs were added and they are well discussed with other references related to the same research area. I think these improvements were necessary and increased the overall quality of the paper.
Congrats to the authors,
This manuscript is a resubmission of an earlier submission. The following is a list of the peer review reports and author responses from that submission.
Round 1
Reviewer 1 Report
This paper describes an interesting approach for assessing the feasibility of executing Very High Frequency Omnidirectional Range (VOR) inspections using Remotely-Piloted Aircrafts (RPA). To this end the authors have developed and implemented a Hardware In the Loop (HIL) platform. This HIL platform has coded using a commercial open code autopilot and integrates the Auto-Pilot system (AP) hardware and the flight simulator, enables the simulated aircraft to be controlled by a radio controller or by an AP. To assess the feasibility of the approach the authors made a series of experiments to verify: 1) if the meet VOR system meets the norms of regulation (the Positioning System (PS) should have the required sample rate, precision and accuracy); 2) if the Autopilot Positioning System (APS) complies with ICAO regulations.
I have following comments that need to be addressed before I recommend this for publication.
Major comments:
1. The abstract is not conclusive and informative. It should be rewritten.
2. In my opinion, a flowchart showing the global approach for assessing the feasibility is missing in section 2. Moreover this section should give an overview of the following sections justifying why they are needed to understand the manuscript as a whole.
3. The conclusions, while not expressive, must be supported by the quantitative results obtained in the previous sections.
Minor comments:
Line 16: give the figures in meters. The error values given in degrees are not expressive and are difficult to visualize.
Line 19. you mean 3.99º?
Line 65. The PS abbreviation is undefined. Please rewrite also this sentence to improve its readability;
Line 65-69. The difference between PS and APS should be smoothly introduced before this paragraph.
Line 195-196: give the values of longitude and latitude in degrees, minutes and seconds. They more readable. The same applies to figure 5.
Line 232: Table 1. The authors should be careful that they are dealing with a circular variable. Therefore they should use the circular statistics for computing the mean and standard variation.
Line 332. Table 5. The authors should be careful that they are dealing with a circular variable. Therefore they should use the circular statistics for computing the mean and standard variation.
Line 429. Table 9. In my opinion the units in degrees are not meaningful. Perhaps they should be expressed in seconds.
Reviewer 2 Report
The article is centered in evaluating the feasibility of using RPAS to perform the VOR system flight inspection. It presents the aircraft and positioning system requirements clearly and develops a trajectory generator to perform the test with less human intervention. The proposed trajectory generator is then tested in a simulator using the autopilot hardware in a HIL scheme.
Dynamic positioning test lack real tests, closer to the intended working conditions, in order to be valid to evaluate the feasibility of using a positioning system.
The simulator model used to evaluate the trajectory generator proposed is not described nor validated and wind or other perturbations that have a huge role into the feasibility of the VOR inspection are not considered.
The errors introduced in the last section should be obtained or estimated using real tests closer to the intended working conditions.